# Methodological Approaches to Talent Identification in Team Sports: A Narrative Review

**DOI:** 10.3390/sports10060081

**Published:** 2022-05-24

**Authors:** Sam Barraclough, Kevin Till, Adam Kerr, Stacey Emmonds

**Affiliations:** 1Carnegie School of Sport, Leeds Beckett University, Leeds LS6 3QS, UK; k.till@leedsbeckett.ac.uk (K.T.); s.emmonds@leedsbeckett.ac.uk (S.E.); 2Leeds United Football Club, Walton Road, Thorp Arch, Leeds LS23 7BA, UK; adam.kerr@leedsunited.com; 3Leeds Rhinos Rugby League Football Club, Bridge Road, Kirkstall, Leeds LS5 3BW, UK

**Keywords:** talent identification, talent development, youth, sport

## Abstract

Talent identification (TID) and talent development (TD) continue to receive significant investment from team sports organisations, highlighting their importance in attempting to identify potential elite athletes. Accompanying this continual pursuit to unearth future talent is an ever-increasing body of research aiming to provide solutions and strategies to optimise TID and TD processes. Therefore, the aim of this review is to provide a summary and critical synthesis of the methodological approaches applied to TID in team sports and present considerations for future TID research. Specifically, this review highlights three key areas for consideration: (1) the timespan of the research design; (2) the use of monodisciplinary or multidisciplinary variables; and (3) the fidelity of the methodological approaches to the assessment of talent. The review highlights the benefits of longitudinal, multidisciplinary, and ecologically valid research designs for TID within team sports.

## 1. Introduction

The lure of success and financial reward is big business for sporting organisations. At the highest levels of team sports, exorbitant investments are made for the recruitment of the most talented athletes who can help organisations accomplish their goals. With this in mind, sporting organisations continue to invest in the identification and development of young talented athletes within their academy systems, with the hope of unearthing or developing potential world class elite athletes of their own [1]. As a product of this investment, youth sport is becoming increasingly professionalised, with organisations now supplying considerable resources for talent identification (TID). TID is defined as “recognising players participating in the sport who have the potential to excel” [2] (p. 1). Once identified, organisations aim to provide appropriate learning environments so that such athletes have the opportunity to realise their potential by maximising the training and development opportunities of prospective talents with the greatest potential for success [3,4]. This occurs through talent development (TD), defined as a “relatively systematic combination of coaching, support, training, and match play designed to progress players” [2] (p. 1).

Due to the popularity and growth of elite athlete development programmes, there is a vast and diverse quantity of TID research available across multiple sports. The variety and depth of such research has been important in establishing an evidence base, providing valuable reference data across sports in multiple disciplines (e.g., technical, tactical, physical, psychosocial), that may be used to distinguish between performance levels. Yet, this volume of research has potentially led to contrasting opinions and widespread misconceptions of talent in high performance team sport settings [5]. For example, it is acknowledged that TID is a complicated process, with the question of “what is talent?” alone proving to be a highly divisive and contradictory topic [6,7]. Due to a lack of consensus on a definition and objective measure of talent, TID (for the purpose of this review) refers to recognising current participants with the potential to progress or to become an elite athlete [2,8]. TID has typically inferred potential based on current performance level [9], yet Bergkamp et al. [10] argued that using performance level as an outcome for TID (i.e., elite vs. non-elite) may be misleading. Performance level is a consequence of one or more (de)selection decisions, and therefore, may only reflect a perception of talented and less talented individuals, rather than an objective measure of talent or potential. Without a clear measure for identifying future elite players, TID has become a significant contributor to research on youth team sport athletes; however, with such a substantial amount of literature, issues emerge relating to the diversity of research methods.

As a key area for research within team sports, several recent review articles [5,10,11,12,13,14,15] raised issues regarding current approaches to TID. These include a lack of longitudinal research designs, the use of monodisciplinary research designs (i.e., physical, psychological, technical, tactical, etc.), and low-fidelity performance characteristics (e.g., isolated sprint testing). However, these methodological issues have yet to be examined collectively, with specific application to team sports as a whole. To date, there are no real consensuses on the underpinning methodology for talent identification or which characteristics of talent may best distinguish athletes which are most likely to progress to the elite level. Researchers and practitioners continue the search for the distinctive characteristics responsible for achieving sporting excellence, but is it fair to say that no consensus may ever be reached, considering the ever-evolving complex and dynamic nature of team sports and the subjective opinions of what constitutes successful performance [16]. Given the ever-expanding volume of interest, research and applied practice surrounding TID, this narrative review aims to provide a summary and critical synthesis of the methodological approaches to talent identification in team sports and to present considerations for future TID research.

## 2. Talent Identification Research Designs

### 2.1. Cross-Sectional Research

Cross-sectional research designs are the most common methodological approach in TID research (i.e., 68% of studies according to a recent scoping review by Baker et al. [11]). Cross-sectional studies often measure specific characteristics within different disciplines (e.g., speed, endurance (physical), passing, dribbling (technical), motivation, confidence (psychological), game intelligence, and general tactics (tactical)) at a one-off timepoint and make comparisons across two or more distinct groups. Previous research has included comparisons of elite vs. non-elite athletes [17], selected vs. non-selected regional athletes [18], academy vs. school athletes [19] or regional vs. national athletes [20]. This type of research is often used to measure the characteristics believed to be linked to successful performance in a cross-section of the sample of interest [21]. Such cross-sectional research designs provide a “snapshot” of performance at a moment in time, which is perhaps indicative of an individual’s expertise or talent.

Cross-sectional study designs have been used in TID across multiple team sports, including soccer [22], rugby union [19], Australian football [23], netball [24], rugby league [25], basketball [26], and field hockey [27]. Whilst this research is of value, the efficacy of cross-sectional designs in identifying talented youth athletes remains in question. For example, research by Gil et al. [28] examined the selection process of a professional soccer club in Spain to identify the physical characteristics of players who were selected into the club’s academy. Players who were selected between the ages of 9 and 10 years were leaner (48.9 mm vs. 66.2 mm sum of skinfolds, *p* < 0.01), quicker (4.96 s vs. 5.53 s in a 30-m sprint test, *p* < 0.001), more agile (5.81 s vs. 6.38 s in a 30-m agility test, *p* < 0.001), jumped higher (29.1 cm vs. 26.9 cm in a countermovement jump test, *p* < 0.01) and possessed greater aerobic endurance (618 m vs. 464 m in the yoyo intermittent recovery level 1 test, *p* < 0.01) than a control group from an open soccer camp who were not selected to train in the club’s academy. If physical advantages at a young age, as observed by Gil et al. [28], are used in TID and selection processes, this seems heavily reliant on the assumption that any physical advantages would remain consistent within individuals across childhood and adolescence, and transfer to adult performance. This fails to account for the influences of individual growth and maturation [29,30,31,32] and the effects of development (i.e., practice, coaching and training) [2]. Similarly, research by Zuber and Conzelmann [33] demonstrated elite youth ice hockey players with higher intrinsic motivation (assessed via 5 motivational constructs–win orientation, goal orientation, hope for success, fear or failure and self-determination), were rated as better players by their coaches (using a 1–100 scale) when judging game performance, in comparison to their less motivated counterparts. Therefore, a key limitation of a cross-sectional research design as a methodological approach is that assessing performance, at a singular time-point, as an indicator of talent, provides limited information on future potential. This is partly due to the non-linear and dynamic nature of development in talented elite youth athletes [34,35], where variables that correlate with a performance advantage at young ages (e.g., an early developing basketball athlete with greater height) may not necessarily be the same factors explaining adult performance or that the individual’s height may be an advantage in adulthood [6]. Research evidence shows the disparate development among youth athletes. For example, a longitudinal case study by Moran et al. [36] displayed substantial fluctuations in academy soccer player’s sprint and jump performances over a 6-year period. Such research confirms that one off performance measures are likely temporary representations of athletic capabilities, where current performance is interpreted as a proxy for potential [9].

In summary, whilst cross-sectional data used in TID is advantageous for comparisons between groups or athletes at a singular timepoint, the inclusion of cross-sectional data in identification or de(selection) decisions within long-term TID/TD programmes can be considered imprudent, as it may prematurely exclude late-developing athletes, given the non-linear development of certain characteristics that may affect performance (e.g., speed, [36]). A more suitable approach is likely to be based on serial measurements of these characteristics over time, to better understand the trajectory of an elite youth team sport athlete’s development [37].

### 2.2. Longitudinal Research

Longitudinal research has been used to follow a cohort of athletes and assess changes in characteristics at two or more time-points [38]. Through taking repeated measurements of an athlete or group of athletes, a longitudinal research design can assess the characteristics that may be linked to performance whilst also assessing changes and development over time [39]. In practice, longitudinal research has greater affinity than cross-sectional research to TD, where regular assessments can serve as a monitoring tool for a group of athletes. Longitudinal research surrounding TID is less common, research that does exist has demonstrated variations in the long-term development of certain characteristics between differing groups, in several sports including rugby league [40], field hockey [41], handball [42], soccer [43], and Australian rules football [44]. Key findings of such studies are summarised in Table 1. Studies were selected as being representative of a variety of team sports, having a minimum of three measurement occasions and a study period of at least 12 months in order to represent longitudinal change between groups that was not attributable to short-term intervention. 

Whilst cross-sectional data can provide differences in characteristics between two distinct groups at singular timepoints, longitudinal research [45,46] provides practitioners with a measure of athlete progression to assess the effectiveness of TID/TD processes [31]. However, one major methodological challenge to longitudinal research is participant dropout, where repeated measures cannot be taken of athletes who are not afforded the opportunity to progress. This is highlighted in the work of Moran et al. [36] who’s final sample of 6 athletes (from an initial 140) were the only individuals to achieve the longevity required for the 6-year period of study on longitudinal monitoring of physical characteristics within a single professional soccer academy. In such cases, a more thorough estimation of sample size requirements that accounts for participant attrition and expected drop out rates may help overcome such methodological challenges.

Most longitudinal research measures change on a group level, possibly sacrificing insight into changes on an individual level, which may provide a more in-depth understanding of development. Through monitoring longitudinal changes in the characteristics that underpin successful performance, researchers and practitioners are likely to be provided with a more valid, continuous indicator of an athlete’s potential to progress based on that athlete frequently achieving the necessary characteristics to be retained within a TD programme. For example, an athlete who progresses through an academy and avoids deselection is likely to possess superior characteristics in one or more disciplines (physical, technical, tactical, psychological) at multiple timepoints, from both an objective (standardised assessments) and/or subjective (coach’s perceptions) perspective, in comparison to their deselected peers. This allows them to continue in the pathway and have an opportunity to reach the professional level in their sport [6], rejecting the notion of TID as a transient process.

### 2.3. Prospective/Retrospective Research Designs

When discussing methodological issues surrounding TID in soccer, Bergkamp et al. [10] stated a key focus of TID research is to evaluate the predictive value of performance characteristics, not just to identify such characteristics. Research has attempted to both prospectively track an athlete’s development into professional status [47], as well as retrospectively examine their development once professional status has been attained [48]. Approaching TID through prospective and retrospective research designs, often leads to TID being conceptualised as a direct relationship between a factor (e.g., height) and adult performance in a particular team sport (e.g., volleyball). For example, research in soccer players who went on to play at international or professional levels as adults, displayed superior performance in several anthropometrical and fitness measures at under 14 to under 16 age groups (i.e., height, body mass, maximal anaerobic power, countermovement jump, 40-m sprint time) [49]. More recent research supports such findings showing that future professional soccer players outperformed their non-professional counterparts in measures of speed (5/10/20-m sprint times), power (countermovement jump height), and endurance (distance covered in yoyo intermittent recovery test level 1) from age ~13/14 years onward [39]. Similar findings have also been shown when investigating psychological [50], tactical [51] and technical [52] characteristics, as well as multidimensional research designs [53]. For instance, Forsman et al. [53], found future elite players outscored non-elite players, at 15 years of age, in tests of dribbling and passing, passing and centering (technical), speed, agility, endurance (physical), motivation (psychological), and “acting in changing situations” (tactical). Whilst these examples of research may aid in establishing characteristics associated with future success (i.e., having better characteristics), research still fails to provide insight into the individual, non-linear developmental patterns of such characteristics [48].

A methodological approach that considers the dynamic nature of TID/TD as a long-term process, whilst also considering future career outcome, allows practitioners and researchers to further understand and examine the relationships and individual developmental trajectories that may influence the future career attainment of the most talented team sport athletes [48]. Studies using such an approach (i.e., longitudinal retrospective) are uncommon in the literature, with some exceptions [38,47,48]. For example, Till et al. [38] retrospectively examined the development of physical characteristics between 13–15 years of age for those players who attained professional, academy and amateur status in rugby league. It was found that the enhanced development of sitting height, speed, change of direction speed and estimated maximal oxygen consumption (VO_2_ max) between 13–15 years of age could differentiate between career attainment outcome of professional and amateur players. Similar findings in soccer [48] showed different patterns of development in tests of vertical jumping and slalom agility when prospectively tracking future professionals and non-professionals, with professionals improving at a faster rate between 12–18 years of age. In contrast, Leyhr et al. [47] found no significant interactions between speed and technical skill development and future adult performance level (i.e., professional vs. non-professional). It should be noted however, inconsistencies in definitions of professional status were observed between the studies, with Leyhr et al. [47] limiting their scope to professional players only within Germany. These contrasting findings potentially suggest a lack of generalisability outside of their respective environments (e.g., sport, country), but also to the wider population due to the restriction in the range present in the respective samples typified by the homogeneity of groups (i.e., selection bias of team sport athletes selected to some form of TID programme [10]). Additionally, the selected studies tended to assess longitudinal development and career attainment interactions at a group level, where a case-by-case individual analysis of players may provide more insight [47].

As such, research designs may aim to identify characteristics important for successful performance, track the fluctuating development of these characteristics through periods of adolescence/maturity, and evaluate their relevance in future career outcomes assessed on an individual level. It should also be noted that due to the complex, myriad of factors responsible for team sport performance, research that is mono-disciplinary in nature (i.e., only examining one component of performance, such as physical characteristics) cannot provide a complete picture of TID. As an extension, research that incorporates an array of potential future successful performance characteristics, and their interactions, into a longitudinal evaluation of the player, appears to be the optimal approach for TID/TD purposes [37].

## 3. A Multidisciplinary Approach

One proposed component of talent is its multi-dimensional nature [6]. Whilst the call for research to adopt a multidisciplinary approach is a recurring message [4,8,13,32,35,54], both current and previous research surrounding TID in team sports has often adopted monodisciplinary designs [19,39,50,51,52]. This was highlighted by an underrepresentation of multidimensional designs in a recent review [5]. The lack of multidimensional designs is perhaps due to the fact that, in reality, the identification of talented individuals is difficult to objectively explain [6]. This is accompanied with the associated methodological challenges of needing to measure variables from each discipline in their entirety [46], combine these into a tool for TID purposes, and implement this across large samples (e.g., nationwide TID processes [55,56]).

Given the challenges of a multidisciplinary research design, a mono-disciplinary design is often utilised. Despite some of the limitations highlighted above, this approach can still provide rich insights for both researchers and practitioners. For example, research from several sports has solely examined physical qualities in relation to TID [57,58,59,60,61]. Additional mono-disciplinary research has shown the value of assessing tactical [51,62], technical [52,63], psychological [33,64], and even genetic traits [65] within TID. In such cases, it may be interpreted that through mono-disciplinary evaluations, an individual’s superiority in one characteristic (e.g., speed) can potentially compensate for weakness in others (e.g., technical/tactical) [35]. Whilst such examples provide a valuable source of information for TID/TD, a mono-disciplinary approach to research, where the outcome variable is related to only one discipline of performance (e.g., physical characteristics), may not fully explain the intricacies of individual talent and development, as it fails to consider “*the interaction of many different elements spinning in the contextual web of final performance*” [66] (p. 2).

The interactions of such elements can also be problematic during the decision-making process for coaches [9]. Namely, the use of multiple sources of information across disciplines in TID decision-making can lead to athlete’s having similar summative scores (across all characteristics) but very different individual performance scores. Figure 1 provides three hypothetical examples of different athlete types where such challenges may occur. In such cases, the decision to de(select) athletes becomes more complex. Here, each athlete has a very similar summative score, creating a choice between those with “the overall package” (even scores across all characteristics—Athlete 1) or those with “something special” (greater scores in specific characteristics—Athlete’s 2 and 3), who’s weaknesses could potentially be masked or substituted by other players within a team sport [9]. As each athlete’s individual profile is unique to them, a multidisciplinary approach allows the identification of an athlete’s ability in various disciplines and characteristics relevant for performance in their sport, whilst also allowing support staff within the environment to evaluate such strengths and weaknesses in order to facilitate a more individualised plan of development [67].

In this regard, a multidisciplinary approach in research to TID may allow for a more holistic profile of youth team sport athletes and increase the utility of TID [23,41,68,69]. Some examples of multidisciplinary research from various team sports are presented in Table 2. 

As highlighted by some of the selected research in Table 2, whilst a multidisciplinary approach is becoming more prominent in TID research within team sports, research within specific disciplines appears more common (e.g., physical). For example, Dimundo et al. [70] utilised seven physical characteristics in comparison to one tactical, when investigating differences in selected and non-selected academy rugby union players, a finding that appears common across selected TID research with physical characteristics more routinely measured [27,45,71]. This is perhaps partly due to the difficulty in assessing some characteristics (e.g., assessing an athlete’s tactical knowledge through retrospective video analysis [70,71]), compared to the ease of assessing others where the application of physical testing batteries and anthropometric measurements are commonplace within TID/TD environments. Where including characteristics from all disciplines in order to provide a balanced, comprehensive approach is not viable, research might aim to evaluate the relative importance of each characteristic relative to their sport. A case study by Jones et al. [72] utilised such an approach, i.e., the perceived importance of various fitness tests from a coach and player’s perspective as a weighting factor for ranking the importance of certain physical qualities for individual players. Again, however, such research is limited to physical discipline, and further research across other disciplines is required.

Despite the multidimensional nature of the studies listed in Table 2, each used a cross-sectional research design [45,70,71,73] or only observed mean performance across two time-points [74], thus failing to understand if the longitudinal development of any of the investigated characteristics influenced TID decisions. Nevertheless, adopting a multidisciplinary approach to TID research appears more valid and applicable to team sports, as team sports require the interaction of multiple characteristics across disciplines [4]. From this perspective it becomes clear that performance in team sports is not synonymous with one set of characteristics from a single discipline, and yet the dominant approach within research is to assess perceived characteristics of relevance within disciplines in isolation [75].

## 4. Signs and Samples

### 4.1. Signs

A large volume of research across various team sports has recognised the multi-disciplinary nature of sports performance, but often in TID research the isolated circumstances in which an athlete’s characteristics are assessed bears little resemblance to performance itself. For example, some predictors of performance in numerous team sports include physical (i.e., speed, strength, and endurance characteristics [19,60,61]), psychological (i.e., achievement motive, motivation, self-confidence and concentration [27,33,73]), technical (dribbling, kicking and shooting [41,71,76]), and tactical (positioning and deciding, pattern recognition [53,64]). Such characteristics are commonly measured in discrete, controlled circumstances such as laboratory or field based-tests in order to obtain reliable and standardised results—a far cry from the open and often chaotic environment in which these characteristics are utilised during team sport performance.

Using a term borrowed from psychology literature, characteristics measured in this way can be termed as “signs” and are said to be conceptually related predictors of the future behaviour or performance of interest [77]. Sign-based tests are said to lack “fidelity” [10], in that they are distinct characteristics measured in a dissimilar task and context to that of the criterion behaviour (team sport performance). For example, assessing speed as a physical characteristic deemed important for differentiating talented and less-talented individuals in terms of their future sport performance using a signs approach may take the form of a 20-metre sprint test (see [22]). Here athletes would be expected to complete multiple trials of a linear sprint, commonly from a stationary start, over a pre-defined distance and with adequate rest-periods to reduce any potential elements of fatigue. In comparison, during actual performance, an athlete would most likely be already moving or adopting a different body position, may need to sprint in a curvilinear fashion and/or include changes of direction and is likely fatigued from prior actions performed. This is then further compounded by the interactions with moving opponents and team-mates, and the perceptual-cognitive and decision-making requirements of such a task. Therefore, a key methodological concern of a signs-based approach is that whilst providing a reliable and valid measure of a specific characteristic for each athlete in that setting, it is clear such an approach lacks resemblance in terms of task and context to how such characteristics would be utilised during on-field team sport performance. In contrast, given the complex, multi-faceted nature of team sport and the inherent difficulty of measuring individual team sport performance, breaking down performance into predictors from various disciplines and investigating their impact on predicting success and future performance makes sense from a practical perspective [10]. Particularly when many of these predictors have been shown to discriminate between performance levels [17,19,70,78,79].

### 4.2. Samples

If performance, skill, or expertise is viewed as the end-goal or outcome (Baker et al. [6]), then it would seem logical for TID research measuring the precursors to these outcomes, to attempt to mimic these criterion behaviours as closely as possible [75]. Such an approach can be termed as “sample” based, in that researchers sample a behaviour in a highly representative context, providing a higher fidelity measure. This sample is more analogous to the criterion (performance) and therefore likely has greater utility in TID for assessing those with greater potential for future performance, particularly in homogenous groups such as team sports [75]. As talent can be viewed as a complex and dynamic construct where future behaviours stem from the combination of psychological, technical, tactical, and physical characteristics [35], a samples approach does appear more valid within TID research in order to investigate how such multidisciplinary characteristics interact and combine to predict or measure actual team sport performance.

Examples of establishing a samples-based approach can be seen from recent research in soccer, where small-sided games (SSGs) have been investigated as potential tools for TID, as they obtain performance under similar task, environmental and behavioural conditions [80,81,82] and have been validated showing moderate-to-large relationships to actual 11 v 11 performance [82]. Fenner et al. [80] investigated player performance in SSGs (subjective scoring of technical aspects rated by the coaches) and match result. There was a significant and large relationship between players judged to have higher technical scores within the SSGs and those found to have more success in SSGs based on an accumulation of points for goals scored and match outcome (r = 0.76, *p* < 0.001). In addition, Bennett et al. [81] showed that higher skilled players (trained within a professional academy) had a significantly greater number of attempted and completed skill involvements in SSGs compared with low-level players (trained within a local academy) (*p* < 0.01). Further research within American Football demonstrated that samples of previous performance, measured via position specific in-game statistics (e.g., percentage pass completion for a quarterback), across a 1-year period at college level, was a statistically significant predictor (*p* < 0.05) of subsequent performance in the National Football League (NFL), whereas signs of performance (i.e., physical tests in the NFL Combine) failed to demonstrate predictive power of future NFL performance [83]. Equally, in Australian Football, O’Connor et al. [64] demonstrated a significant difference in recent match-play performance (sample) between selected and non-selected athletes into a national programme (*p* < 0.001). Recent match performance was also identified as a predictor variable that could discriminate between selected and non-selected, with a large standardised coefficient (0.851), indicating its importance. It should be noted however, that recent match performance in this study was based upon a coded variable indicating selection for participation in regional camps and tournaments and thus this sample of behaviour may reflect perceived match performance as opposed to actual performance.

### 4.3. Subjective Expert Opinion

Given the complexity of sampling performance in its entirety, one method utilised in order to provide a samples-based assessment is the inclusion of a subjective expert opinion (SEO), where a coach or practitioner can provide a holistic rating of player performance (e.g., a score from 1 to 4, [84]. Research has shown that inclusion of subjective ratings from coaches improves predictive models within TID in comparison to objective data alone [55,74]. However, the basis of and validity of such ratings is yet to be established with research showing a lack of agreement between coaches [85], an inability for coaches to accurately rate performance within specific disciplines (e.g., physical, [86,87]) and suggestions that ratings are potentially biased [82] and could be based on a coaches’ perceived ability to influence and develop a player rather than solely on athlete ability alone [88,89]. Evidence of such biases has shown subjective ratings may vary based on an individual’s stage of maturation and rate of growth, with a trend for ratings to decline for players around the time of their growth spurt, before increasing again post growth spurt [84]. Equally, it may be expected that maturity timing (e.g., late vs. early) may influence coach ratings, as early maturing players typically have physical advantages in size, strength, and speed versus their less mature counterparts [90]. In such scenarios, a samples approach where individuals are grouped relative to their biological age (i.e., “bio-banding”) may remove such physical biases, allowing later maturing players more opportunity to exhibit their tactical and technical proficiency [91], potentially facilitating a more valid sample of performance through SEO. Due to the lack of evidence on the validity and reliability of SEO’s, there are concerns regarding the use of coach ratings alone, as they may lack a shared and explicit criterion upon which ratings are based. Given such information, TID should attempt to utilise both objective and subjective profiling information to help inform their decision-making processes rather than solely rely on clinical judgement [75].

## 5. Conclusions

The current review highlights three key methodological approaches relevant to TID research, namely, the time-course of the research design (i.e., cross-sectional or longitudinal; prospective or retrospective), the disciplines of interest (i.e., mono or multi-disciplinary designs), and the assessment method applied (i.e., signs or samples). These methodological approaches have a range of strengths and limitations regarding TID research, and remain pertinent within research related to any team sport due to the dynamic, multidimensional, and complex demands of such sports.

To summarise, cross-sectional designs fail to account for the non-linear development of youth athletes and the emergenic, dynamic and symbiotic conceptualisation of talent [6]. This may potentially lead to misrepresentations of an individual’s potential when undertaking (de)selection decisions, as different characteristics will evolve and develop at different rates for each individual athlete, in conjunction with the potentially confounding effects of growth, maturation, and development [31,61,92]. With this in mind, it is proposed that a longitudinal approach to TID research may be more beneficial, as it may provide insights into the individual developmental changes of indicators of talent and their effect on (de)selection decisions.

Equally, although team sports are complex, dynamic, and multi-dimensional in nature [74], TID research is often monodisciplinary. This is perhaps due, in part, to the relative ease of examining certain characteristics (i.e., anthropometric and physical characteristics) which are often routinely measured within embedded TID programmes (i.e., pre-season testing). In this regard, a multidisciplinary approach to TID is recommended to provide a more holistic evaluation of an athlete, accounting for their strengths and weaknesses in multiple aspects of performance, which can further facilitate TD and (de)selection processes.

Finally, the context in which indicators of TID are measured must be questioned. Discrete and controlled tests (“signs”), whether conducted in the laboratory or field, lack ecological validity and transference to actual performance within team sports. Accordingly, a samples-based approach may be more appropriate in TID programmes, where judgements are made based on assessments that more closely mimic the context, environment, and task of team sport performance [23,64,81,83], including the subjective expert opinions of relevant staff [84,86,93].

## 6. Directions for Future Research

Regardless of the sport, TID is and will remain a key area of interest within both research and practice. Despite the plethora of methodological approaches, the current review highlights and reinforces some key considerations for future research:Future TID research should strive to adopt a longitudinal research design in order to provide regular and comprehensive evaluations of athlete’s performance in relevant characteristics and their individual rates of change as possible indicators of potential.A multidisciplinary approach to research would allow for more comprehensive athlete profiling and serve not only as a potential tool for TID but to also augment TD processes within team sport environments.Investigating both objective and subjective data through a combined approach of signs, samples and subjective expert opinions would allow researchers to bridge the gap between relevant characteristics and their transfer to performance, with an added perspective from “the coach’s eye”.

## 7. Practical Applications

As well as providing recommendations for future research, these methodological considerations should also serve as a comprehensive framework to athlete profiling, thus informing TID, TD and talent selection processes.

A comprehensive approach to athlete profiling should:Identify key actions for successful match play and the underpinning multidisciplinary characteristics required to perform such actions.Profile the actions and relevant characteristics through multiple methods—signs, samples, and subjective expert opinions.Repeat the profiling longitudinally to account for non-linear development whilst also examining the trend of development as an indicator of potential i.e., showing the capacity to successfully perform such actions in the future.

## Figures and Tables

**Figure 1 sports-10-00081-f001:**
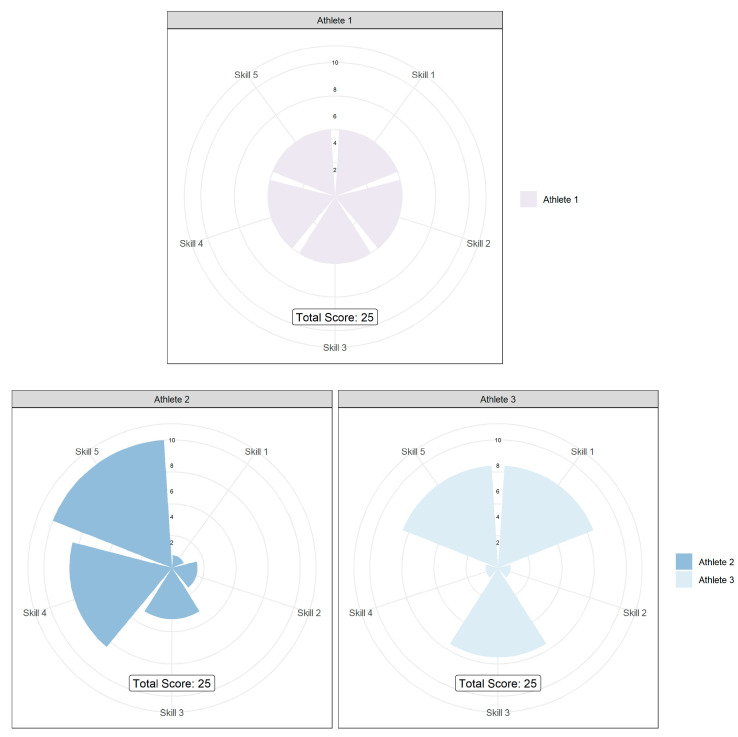
Hypothetical performance comparison for 3 athletes.

**Table 1 sports-10-00081-t001:** Examples of Longitudinal Research for TID in Team Sports.

Authors/Sport	Sample/Timeframe	Objectives	Key Findings
Till et al., 2013 [40]/rugby league	81 male junior rugby league players from under 13-under 15/3 consecutive years.	Compare longitudinal development of physical and anthropometric characteristics considering position and selection level in junior rugby league players.	1. Selection level (national vs. regional) had a significant overall main effect on physical and anthropometric characteristics. 2. Players who moved up in selection level significantly improved sprint speed and were the quickest at under 15 age category. 3. There was a significant interaction between maturation and time for sprint speed, vertical jump, and medicine ball throw.
Matthys et al., 2013 [42]/handball	94 youth handball players from under 14-under 18/3 consecutive seasons.	Assess longitudinal changes in anthropometry and physical performance between elite and non-elite handball players.	1. Elite players did not improve their physical performance more rapidly than non-elites and had similar anthropometric profiles. 2. Elite players performed significantly better on the intermittent endurance, speed, and coordination items. It was revealed Yo-Yo distance and coordination with and without ball discriminated most between the two playing levels.
Roescher et al., 2010 [43]/soccer	130 male youth soccer players aged under 14-under 18/5 consecutive years with the exception of 1 year.	Investigate the development of intermittent endurance capacity, the underlying mechanisms affecting this development and attained adult playing level in talented youth soccer players.	1. From 15 years of age players who reach professional status show a faster development pattern than non-professionals. 2. Both hours spent in soccer-specific training and hours spent in additional training were positively related to the development of intermittent endurance capacity.
Elferink-Gemser et al., 2007 [41]/field hockey	30 elite and 35 sub-elite male and female youth field hockey players from under 14-under 16/3 consecutive years.	Identify the performance characteristics that may help identify future elite hockey players.	1. Both male and female elite players scored better than sub-elite on technical and tactical variables. 2. Female elite players also scored better on interval endurance capacity, motivation, and confidence. 3. Male and female elite players improved more than their sub-elite counterparts on interval endurance capacity and slalom dribble across the study period.
Pyne et al., 2005 [44] /Australian rules football	283 Australian rules football players/3 consecutive years.	Determine the relationships between anthropometrics and physical fitness tests and subsequent career progression.	1. Drafted players were faster (5, 10 and 20-m), had higher estimated VO2 max and a faster agility run performance than non-drafted players. 2. No substantial differences in anthropometric or jump tests were found between drafted and non-drafted players.

**Table 2 sports-10-00081-t002:** Examples of Multidisciplinary TID Research.

Authors/Sport	Sample	Variables	Disciplines	Key Findings
Dimundo et al., 2021 [70]/Rugby Union	74 elite under 15 male youth rugby union players.	Height, body mass, 10-m and 20-m sprint time, counter-movement jump, isometric hip extension, dominant handgrip strength, date of birth, perceptual-cognitive video simulation.	Physical, tactical	1. Selected players to an academy outperformed those not selected in body mass, handgrip strength, isometric hip extension and 20-m sprint (*p* < 0.05). 2. No significant differences were apparent for the perceptual-cognitive test.
(Elferink-Gemser et al., 2004) [27]/Field Hockey	126 elite male and female youth field hockey players aged 11–16.	Height, body mass, percentage body fat, peak shuttle sprint, repeated shuttle sprint, slalom sprint performance, interval endurance capacity, peak shuttle dribble, repeated shuttle dribble, slalom dribble, general tactics, tactics for possession of the ball, tactics for non-possession of the ball, motivation, confidence, anxiety control, mental preparation, team emphasis and concentration.	Physical, technical, tactical, psychological	1. Stepwise discriminant analysis predicted better tactics for possession of the ball, being younger, having a higher motivation, and a quicker slalom dribble could best discriminate between elite and sub-elite players. 2. Elite youth players scored better than sub-elite youth players on technical (peak dribble and repeated dribble shuttle run), tactical (general tactics, tactics for possession and non-possession of the ball) and psychological variables (motivation) (*p* < 0.05).
Falk et al., 2004 [45]/Water Polo	24 male youth water polo players aged 12–14.	50, 100, 200 and 400-m freestyle swim, 100-m breaststroke, 100-m butterfly, 50-m dribbling, throwing at goal, throwing for distance, vertical jump from water, game intelligence.	Physical, technical, tactical	1. Two years before selection to the junior national team, players who were selected outperformed those non-selected on game-intelligence, 50-m dribbling and all swim tasks except 50-m freestyle and 100-m breaststroke (*p* < 0.05). 2. Using an average rank score, predictions for 67% of players were in agreement with final selections.
Sieghartsleitner et al., 2019/Soccer	117 elite under 14 youth soccer players.	Age, relative age, age at peak height velocity, height, body mass, in-game performance, YoYo intermittent recovery test level 1, 40-m sprint, agility, dribbling, passing, juggling, achievement motive, achievement goal orientation, self-determination, importance of football within family, parent’s priority of sport vs. school, financial investment, time investment, practice and play up to age 12.	Physical, technical, psychological, sociological	1. A holistic model combining all predictor variables had the greatest accuracy (88%) in correctly predicting who would achieve professional vs. non-professional status 5 years later.
Woods et al., 2015 [71]/Australian Rules Football	84 elite under 18 Australian rules football (AF) athletes.	Standing height, dynamic vertical jump height on non-dominant leg, 20-m multistage fitness test, kicking, handballing, video decision-making.	Physical, technical, tactical	1. Those selected for state representation (“talent identified”) outperformed non-talent identified on each test (*p* < 0.05). 2. Using a summative score receiver operating characteristics were able to correctly classify 95% of talent identified and 86% non-talent identified participants (AUC = 95.4%).

## Data Availability

Not applicable.

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
