# Peer review of "Methodological Approaches to Talent Identification in Team Sports: A Narrative Review"

_sports, 2022, doi:10.3390/sports10060081_

Round 1

Reviewer 1 Report

  1. What does p.1 (line 32, 35) mean?
  2. Line 32 missing dot after ref [1].
  3. Maybe the authors should have pointed out the role of genetic tests in identifying the potential of young athletes.

Author Response

Thankyou for your thorough review of our manuscript, we appreciate the time taken to conduct such a review. We have considered your comments and suggestions and made changes to the paper accordingly. These changes are summarised below.

  1. Line 32-35 refers to the increasing professionalisation of youth sport due to the investment in Talent Identification. Talent Identification is then defined. - This sentence has been amended for further clarity.
  2. Punctuation error has been addressed.
  3. The role of genetic testing in identifying potential has now been highlighted, please refer to paragraph 2 in multidisciplinary approach section.

Reviewer 2 Report

A well-written article, an interesting topic.

The tables are illegible (opaque).

In conclusion, I would expect specific springs for sports organizations or. selected sports on how to work better and evaluate TID.

Figure 2 is very simple, even unnecessary, text would suffice. I do not understand why picture 2 is in front of the picture 1, respectively picture1 is 2x.

Author Response

Thankyou for your thorough review of our manuscript, we appreciate the time taken to conduct such a review. We have considered your comments and suggestions and made changes to the paper accordingly. These changes are summarised below.

1. Thankyou.

2. Table formatting in the pdf version of the manuscript has been conducted by the journal, tables should not be transparent and overlapping page headers.

3. Respectfully, following the conclusion is a practical applications section that aims to highlight how to potentially work better and evaluate TID. Please refer to page 13, line 467.

4. Figure 2 has been removed as suggested. Figure 1 should not have been included 2x, again with manuscript formatting being conducted by the journal.
